



# Evaluation of the Chemical Composition of Gas and Particle Phase Products of Aromatic Oxidation

Archit Mehra[1], Yuwei Wang[2], Jordan E. Krechmer[3], Andrew Lambe[3], Francesca Majluf[3], Melissa A. Morris[3, ^], Michael Priestley[1*], Thomas J. Bannan[1], Daniel J. Bryant[4], Kelly L. Pereira[4], Jacqueline F. Hamilton[4], Andrew R. Rickard[4,5], Mike J. Newland[4], Harald Stark[3] Phil Croteau[3], John T. Jayne[3], Douglas R. Worsnop[3], Manjula R. Canagaratna[3], Lin Wang[2] and Hugh Coe[2,1]

[1]Centre for Atmospheric Science, School of Earth and Environmental Sciences, The University of Manchester, Manchester, M13 9PL, UK

[2]Shanghai Key Laboratory of Atmospheric Particle Pollution and Prevention (LAP3), Department of Environmental Science & Engineering, Jiangwan Campus, Fudan University, Shanghai 200438, China

[3]Center for Aerosol and Cloud Chemistry, Aerodyne Research Inc, Billerica, Massachusetts, USA

[4]Wolfson Atmospheric Chemistry Laboratories, Department of Chemistry, The University of York, York, UK

[5]National Centre for Atmospheric Science (NCAS), University of York, York, UK

*Now at Department of Chemistry and Molecular Biology, University of Gothenburg, Gothenburg, Sweden

^Department of Chemistry and Cooperative Institute for Research in Environmental Sciences (CIRES), University of Colorado Boulder, Boulder, CO, USA

*Correspondence* to Hugh Coe (hugh.coe@manchester.ac.uk)

**Abstract**

Aromatic volatile organic compounds (VOC) are key anthropogenic pollutants emitted to the atmosphere and are important for both ozone and secondary organic aerosol (SOA) formation in urban areas. Recent studies have indicated that aromatic hydrocarbons may follow previously unknown oxidation chemistry pathways, including autoxidation that can lead to the formation of highly oxidised products. In this study we evaluate the gas and particle phase ions formed during the hydroxyl radical oxidation of substituted $C_9$-aromatic isomers (1,3,5-trimethyl benzene, 1,2,4-trimethyl benzene, propyl benzene and isopropyl benzene) and a substituted polyaromatic hydrocarbon (1-methyl naphthalene) under low and medium NOx conditions.

The majority of product signal in both gas and particle phases comes from ions which are common to all precursors, though signal distributions are distinct for different VOCs. Gas and particle phase composition are distinct from one another, and comparison with the near explicit gas phase Master Chemical Mechanism (MCMv3.3.1) highlights a range of missing highly oxidised products in the pathways.

In the particle phase, the bulk of product signal from all precursors comes from ring scission ions, many of which have undergone further oxidation to form HOMs. Under perturbation of OH oxidation with increased $NO_x$, the contribution of HOM ion signals to the particle phase signal remains elevated for more substituted aromatic precursors. Up to 25 % of product signal comes from ring-retaining ions including highly oxygenated organic molecules (HOMs); this is most important for the more substituted aromatics. Unique products are a minor component in these systems, and many of the dominant ions have ion formulae concurrent with other systems, highlighting the challenges in utilising marker ions for SOA.

## 1    Introduction

Volatile Organic Compounds (VOCs) are emitted from both natural and anthropogenic sources and their oxidation in the troposphere is important for reactive chemistry leading to ozone (Atkinson, 2000; Derwent et al., 1998) and secondary organic aerosol (SOA) formation(Ziemann et al., 2012). This can have severe air quality, environmental and health impacts (Hallquist et al., 2009). Globally, VOCs such as isoprene have been estimated to be the largest contributors to SOA formation (Guenther et al., 1995; Simpson et al., 1999). However, in urban and industrialised areas, other sources of VOC become increasingly important (Borbon et al., 2013; Karl et al., 2009; Liu et al., 2008b; McDonald et al., 2018). Aromatics are one such class of VOC, which are emitted from fuel use, biomass burning, solvent use and industry (Corrêa and Arbilla, 2006; Liu et al., 2008a).





In the troposphere aromatic hydrocarbons primarily react with the hydroxyl radical (OH)(Calvert, 2002). Detailed mechanisms
of aromatic oxidation have been generated previously from experimental work in simulation chambers(Calvert, 2002) and this
chemistry has been included in chemical mechanisms such as the Master Chemical Mechanism (MCM: mcm.york.ac.uk) and
SAPRC whose primary goal is to describe ozone formation (Bloss et al., 2005; Carter and Heo, 2013; Metzger et al., 2008;
Suh et al., 2003; Volkamer et al., 2002). The aromatic oxidation mechanism in the MCM has not been updated  since 2005
(Jenkin et al., 2003, Bloss et al., 2005) and this can only be achieved once sufficient evidence of the rate, branching ratios and
product distributions has been obtained. Explicit chemical mechanisms often lead to large discrepancies between modelled
and measured SOA formation (Johnson et al., 2006; Khan et al., 2017; Volkamer et al., 2006), which may be associated with
both uncertainty in the formation pathways of semi-volatile and low volatility species and a poor representation of aerosol
volatility and partitioning.

SOA is made up of thousands of oxidised organic compounds which each exist at very low, often sub-ppt levels in the
atmosphere (Hallquist et al., 2009).  This makes measurement of generated products challenging even under controlled
laboratory conditions. However, the increasing availability of high-resolution instrumentation that enable real time detection
of thousands of molecules (Hallquist et al., 2009) has allowed more detailed chemical characterisation of the gas and particle
phase VOC oxidation products. Chemical ionisation is extremely powerful in this area due to soft ionisation schemes, such as
$I^-$, $NO_3^-$ and $H_3O^+$, that form clusters with the types of molecules that sit within atmospherically relevant oxidation ranges
(Isaacman-Vanwertz et al., 2018; Isaacman-VanWertz et al., 2017). These techniques have enabled identification of many new
oxidation products, thereby supporting more detailed mechanistic studies (Molteni et al., 2018; Wang et al., 2015).

Recent mechanistic studies have highlighted the importance of previously unknown pathways, such as autoxidation (Molteni
et al., 2018; Wang et al., 2017) and multi-generational OH-attack(Garmash et al., 2020; Zaytsev et al., 2019), for the formation
of highly oxygenated organic molecules (HOM) which may contribute to new particle formation (NPF) and rapid SOA
formation and growth. Recent work has shown that autoxidation may play a major role on SOA formation at regional and
global scales due to the potential for autoxidation of aromatics to continue to be favourable even under higher NO conditions
(Crounse et al., 2013; Pye et al., 2019). Furthermore, reductions in $NO_x$ in the US and Europe are enabling autoxidation to
play an increasing role under urban conditions (Praske et al., 2018). In order to ascertain the importance of aromatic SOA,
detailed laboratory characterisation of SOA composition and aromatic oxidation mechanisms for a broad suite of relevant
aromatics is required alongside ambient measurements.

Gas phase chemistry and SOA formation has been well studied for the most ubiquitous aromatic VOCs; benzene and toluene
(Bruns et al., 2016; Molteni et al., 2018; Ng et al., 2007; Suh et al., 2003; Volkamer et al., 2002). However, more recently the
importance of substituted aromatic hydrocarbons in urban areas has been highlighted for both ozone and SOA formation
(Kansal, 2009; Monod et al., 2001). Furthermore, substituted aromatics have high OH reactivities and aerosol mass yields
(Odum et al., 1996) so despite existing at generally smaller concentrations than their less substituted homologues, reductions
in their emissions may be essential to improving urban air quality  (Von Schneidemesser et al., 2010). Uncertainty exists in
the emission inventories of $C_7$-$C_9$ aromatics and this could be important when considering the differences between ozone and
SOA formation in developed and developing megacities, particular with a lack of speciation of aromatics in fuels globally
(Borbon et al., 2013). Recent work has begun to fill this gap in the chemical knowledge, and it can be seen that though other
pollutants have been decreased in newer fuels, an increasing trend is observed in the levels of aromatics in cities such as Beijing
(Peng et al., 2017). In urban environments, the impact of this is not easy to constrain, particularly as these emissions can have
complex interactions with biogenic VOCs (Kari et al., 2019) impacting upon SOA composition and yields(McFiggans et al.,
2019).

$C_9$ aromatics are of particular interest in urban areas, with major emissions sources including   unburnt exhaust
emissions(Corrêa and Arbilla, 2006; Na et al., 2005), evaporative losses (Miracolo et al., 2012; Rubin et al., 2006) and solvent
use(Zhang et al., 2013). Another group of aromatics which are increasingly a focus of research are polyaromatic hydrocarbons
(PAH) which can be emitted from vehicular emissions (Miguel, Kirchstetter and Harley, 1998) and biomass burning (Bruns
et al., 2016). With < 50 % of SOA products identified (Hamilton et al., 2005; Sato et al., 2007), it is important to build up a
detailed chemical insight in order to improve mechanistic understanding of which chemical pathways are leading to SOA
formation(Atkinson and Arey, 2007). Previous studies of HOMs  have focused on either very different precursors(Mentel et
al., 2015) or a small range with limited comparison between isomers(Molteni et al., 2018).

In this study we focus on the photochemical oxidation of substituted aromatic hydrocarbons which have recently been found
to form HOMs through rapid intramolecular autoxidation reactions(Wang et al., 2017). Though VOCs are often arbitrarily
grouped in models(Carter, 2000; Yarwood et al., 2005), recent work has found that that the location, number and isomeric
structure of substituent groups on the benzene ring can have implications on SOA yield, chemical composition and physical
properties(Li et al., 2016). In this study we investigate a range of isomers of $C_9$-aromatics to evaluate this effect in more detail.
Of the aromatics studied in these experiments, molecular understanding of SOA products is most well developed for the
trimethyl benzene isomers (Huang et al., 2014; Li and Wang, 2014; Sato et al., 2012, 2019). Though other aromatics have been
studied, these have either lacked molecular level detail or focused on gas phase oxidation and lacked detailed characterisation
of SOA (Chan et al., 2009; Li et al., 2016; Molteni et al., 2018; Ping, 2013). Here we present a detailed characterisation of the
chemical composition of the oxidation products of a range of substituted aromatics (isomers of $C_3$-substituted aromatics and a


PAH): propyl benzene (PROPBENZ), isopropyl benzene (IPRBENZ), 1,3,5-trimethyl benzene (TMB135), 1,2,4-trimethyl benzene (TMB124), and a substituted PAH, 1-methyl naphthalene (METHNAP). We evaluate the trends in chemistry across different isomers under low and medium NOx conditions and discuss the relative importance of different oxidation pathways.

## 2   Methodology

### 2.1   Oxidation of VOCs

We conduct our experiments in an Aerodyne Potential Aerosol Mass (PAM) oxidation flow reactor (OFR) (Lambe et al., 2011) affording short experimental timescales and the ability to generate consistent and reproducible oxidation conditions. Aromatic VOCs were injected into a carrier gas of synthetic air through use of an automated syringe pump, either neat or diluted in carbon tetrachloride ($CCl_4$). In the OFR, OH, $HO_2$ and NO radicals were generated via $O_2$ + $H_2O$ + $N_2O$ photolysis at 254 and 185 nm via the following reactions:

$$O_2 + hv_{185} \rightarrow 2O$$

$$H_2O + O_2 + hv_{185} \rightarrow OH + HO_2$$

$$N_2O + hv_{185} \rightarrow O(^1D) + N_2$$

$$O + O_2 \rightarrow O_3$$

$$O_3 + hv_{254} \rightarrow O(^1D) + O_2$$

$$O(^1D) + H_2O \rightarrow 2OH$$

$$O(^1D) + N_2O \rightarrow 2NO$$

All experiments were carried out at a temperature of 26 ℃, relative humidity of 35 % and constant gas flow of 10 SLM through the OFR, including injection of ~3 % $N_2O$ at the inlet to generate NO in a subset of experiments(Lambe et al., 2017; Peng et al., 2018). At these conditions, the estimated OH exposure in the OFR were in the range of $(1.5-1.7)E^{12}$ molecules $cm^{-3}$ $s^{-1}$. These elevated exposures are not attainable in conventional environmental chambers and may prove relevant for understanding photochemistry in urban areas with high oxidation capacities such as those observed in Chinese megacities(Tan et al., 2019). In experiments where $N_2O$ was added to the OFR, the NO:$HO_2$ concentration ratio was approximately 0.5 as calculated using an adapted version of the OFR photochemical box model described in (Li et al., 2015) and (Peng et al., 2015). Between experiments the OFR was flushed with humidified synthetic air at full lamp power for 12-36 hours until the particle mass generated was reduced to background concentrations, measured by a scanning mobility particle sizer (SMPS) and an aerosol mass spectrometer (AMS). During this time the walls were cleaned of semi-volatiles and precursors, measured by I-CIMS and Vocus PTR-MS. Backgrounds were determined under lights off/on, with/without precursor injection conditions and spectra recorded with all of the instrumentation. The conditions discussed herein are low NOx ([NO] < 0.1 ppb) and medium $NO_x$ ([NO] > 1 ppb, [NO]: [$HO_2$] = 0.5).

| Precursor | isopropyl benzene | | 1-methyl naphthalene | | propyl benzene | | 1,2,4-trimethylbenzene | | 1,3,5-trimethylbenzene | |
|---|---|---|---|---|---|---|---|---|---|---|
| Condition | Low-NOx | Medium-NOx | Low-NOx | Medium-NOx | Low-NOx | Medium-NOx | Low-NOx | Medium-NOx | Low-NOx | Medium-NOx |
| VOC mixing ratio (ppb) | 245 | 510 | 57 | 58 | 221 | 230 | 237 | 245 | 470 | 485 |

Table 1 - VOC concentrations during all experiments

### 2.2   Measurements

#### 2.2.1   FIGAERO-I-CIMS

A time of fight chemical ionisation mass spectrometer (Lee et al., 2014) using an iodide ionisation system was coupled with a filter inlet for gases and aerosols (FIGAERO) (Lopez-Hilfiker et al., 2014) for detection of particle phase composition (I-CIMS herein). Particle mass concentrations were monitored using a TSI Scanning Mobility Particle Sizer and collection time on the FIGAERO filter was varied to ensure comparable mass of aerosol in each sample for the different precursors. The FIGAERO



thermal desorption cycle consisted of a 15 minute temperature ramp to 200 °C, held at that temperature for 10 minutes and
140 then cooled down over 15 minutes.

The gas phase inlet consisted of a piece of 0.5 m length ¼" I.D. PFA tubing from which the I-CIMS sub-sampled 2 slpm. The
aerosol phase inlet consisted of 0.5 m stainless steel through which 2 slpm was pulled over a Teflon filter. I$^-$ reagent ion was
made by flowing N2 over a permeation tube containing methyl iodide($CH_3I$), followed by ionisation through a Po-210
ionisation source. This flow enters an ion molecule reaction region (IMR) which was maintained at a pressure of 100 mbar
using an SSH-112 pump fitted with a pressure controller. The IMR pressure was automatically maintained at a set value using
an actuated valve connected to the pump line.

### 2.2.2 Vocus-PTR

A Vocus proton-transfer-reaction time of flight mass spectrometer (PTR-TOFMS; Vocus hereafter) was used in this
experiment to measure gas-phase organic compounds (Krechmer et al., 2018). Equipped with a newly-designed focusing ion-
150 molecule reactor (FIMR), the Vocus was able to measure organics with a wide range of volatilities (Riva et al., 2019). An SSQ
(short-segmented quadrupole) pressure of 2.0 mbar and an axial voltage of 420V were used, corresponding to an E/N ratio of
~ 100 Td, which was lower than typical to reduce fragmentation of labile SVOCs (Krechmer et al., 2018). A total sample flow
of 2.2 LPM was maintained by a pump to minimise delay times due to the inlet (Pagonis et al., 2017), of which approximately
125 sccm was sampled into the FIMR through the PEEK tube. To reduce inlet blockages by the high mass loadings of aerosols
from the OFR, a filter was connected just before the entrance of the Vocus, which may have reduced transmission of some
SVOCs and LVOCs. Background checks by injection of clean air and calibrations by injection from a multi-component
standard cylinder (Apel-Riemer environmental) were performed every 2 hours in this experiment. With a resolving power of
12000 Th/Th at 200 Th, molecular formula could be assigned to most of detected ions with a mass accuracy better than 3ppm.

### 2.2.3 Orbitrap LC-MS

Ultra-performance liquid chromatography ultra-high resolution mass spectrometry (Dionex 3000, Orbitrap QExactive,
ThermoFisher Scientific) was used for sample analysis. Compound separation was achieved using a reverse-phase C18 column
(Accucore, ThermoFisher Scientific) with the following dimensions: 100 mm (length) × 2.1 mm (width) and 2.6 μm particle
size. The column was heated to 40 °C during analysis. The solvent composition consisted of water with 0.1 % (v/v) of formic
acid (A) and methanol (B) (optima LC-MS grade, ThermoFisher Scientific). Gradient elution was used, starting at 90 % (A)
165 with a 1 minute post-injection hold, decreasing to 10 % (A) at 26 minutes, returning to the starting mobile phase conditions at
28 minutes, with a 2 minute hold to re-equilibrate the column (total run time = 30 minutes). The flow rate was set to 0.3 ml/min
with a sample injection volume of 2 μl. Samples were stored in a temperature controlled autosampler tray during analysis
which was set to 4 °C. The mass spectrometer was operated in negative and positive ionisation mode with a scan range of m/z
85 to 750. Heated electrospray ionisation was used, with the following parameters: capillary and auxiliary gas temperature of
320 °C, sheath gas flow rate of 70 (arb.) and auxiliary gas flow rate of 3 (arb.). Tandem mass spectrometry was performed
using higher energy collision dissociation with a normalised collision energy of 65, 115. Data was analysed using Compound
Discoverer version 2.1 (ThermoFisher Scientific). Full details regarding the data processing methodology can be found in
Pereira et. al (2020) (under-review in ES&T). Briefly, molecular formulae assignments were allowed unlimited C, H, O atoms,
up to 2 S atoms and 5 N atoms, plus > 2 Na atoms and 1 K atom in positive ionisation mode. Only compounds with a mass
error < 3 ppm and signal-to-noise ratio > 3, hydrogen-to-carbon ratio of 0.5 to 3 and oxygen-to-carbon ratio of 0.05 to 2 were
included in the data set. Instrument artefacts were removed from the sample data if the sample/artefact peak area ratio > 3.

### 2.3 Data Analysis

Data analysis of I-CIMS and Vocus was performed using the "Tofware" package (version 3.1.0) running in the Igor Pro
(WaveMetrics, OR, USA) environment(Stark et al., 2015). Time of flight values were converted to mass-to-charge ratios in
the I-CIMS using I-, I.$H_2O^-$ and $I_3^-$. The instrument was operated at a ~ 8000 Th/Th resolving power. Several impurity ions,
ions from ubiquitous VOCs, and ions from oxidation of products were used for the mass calibration of Vocus, which are all
clear and unique peaks in the mass spectrum, including $C_2H_5O^+$ (acetaldehyde, 45.033491 Th), $C_2H_7O_3^+$ (hydrate ion of acetic
acid, 79.038971 Th), $C_6H_9O_3^+$ (common oxidation product of all precursors in our experiments, 129.054621 Th), $C_8H_{11}O_2^+$
(common oxidation product of all precursors in our experiments, 139.075356 Th), and $C_{10}H_{31}O_5Si_5^+$
(decamethylcyclopentasiloxane (D5, contaminations from personal care products(Coggon et al., 2018)), 371.101233 Th).
Further analysis was carried out in custom Python 3 procedures using the packages Pandas, Matplotlib and Numpy.

In this work, the chemical composition of the gas phase species generated in all the experiments is characterised by the Vocus.
While the FIGAERO is capable of providing information about both gas and particle phases, the gas phase I-CIMS spectra
obtained during the medium NOx experiments was complicated by the presence of high nitric acid formed in the OFR from
the N2O precursor(Lambe et al., 2017; Peng et al., 2018). High nitric acid depletes the I$^-$ reagent ion to form $NO_3^-$, which
subsequently acts as an additional reagent ion and complicates interpretation of the observed CIMS spectrum. Thus, the gas
phase FIGAERO measurements were deemed unsuitable for the comparison of medium and low NOx conditions and the HR





analysis of the I-CIMS spectra has been carried out only for the particle phase I-CIMS FIGAERO data obtained in all the experiments, for which nitric acid levels were lower, resulting in pure iodide reagent ion chemistry. While this setup provides
thermal desorption profiles which have been previously used to draw conclusions about the volatility distribution of ions(Lopez-Hilfiker et al., 2016b; Schobesberger et al., 2018; Stark et al., 2017), in this study we instead integrate the desorption profiles in order to compare the overall composition of the different precursors.

The observed mass spectra were first mass calibrated (10 ppm mass accuracy) and then the observed high resolution peaks were fit using a multi-peak fitting algorithm. The exact mass of the multiple peaks are then matched with the most likely
elemental formula. A consistent approach was taken for the high resolution (HR) peak identification in all experiments.  To eliminate bias, separate peak lists were generated for each precursor and experimental condition. Mass ranges were selected in order to focus on the more oxidised products: m/z 200-500 for I-CIMS and m/z 75-300 for Vocus (except mz 79-81, 117-123 for all precursors, and 143-144 only for methylnaphthalene, the range dominated by the signals of precursor, solvent, the hydrated ion of acetic acid, and their isotopes). The mass spectra in this range were fitted with peaks in order of descending
signal contribution until the signal-to-noise made peak identification not possible. Predicted oxidation products for each aromatic precursor were obtained from the MCM v3.3.1 using the "extract" function on the web portal (mcm.york.ac.uk). Fitted unknown peaks were firstly compared with this list of MCM products and assigned through use of the iterative peak assignment method which assigns unknown peaks to elemental formulae in a reference list if they are in the correct position (Stark et al., 2015).

Quantification of highly oxidised species measured by I-CIMS is challenging due to the lack of availability of standards for many of the observed products. Previous attempts at quantification have used functional group dependencies, collision limit sensitivities or those derived from ion-adduct declustering scans (Lopez-Hilfiker et al., 2016a). Experimental limitations exist in the use of these techniques, meaning that quantification remains a challenge. For PTR, it has previously been reported that the $H_3O^+$ capture rate and therefore the sensitivity of PTR to all ions lies within a very narrow distribution, unlike iodide
sensitivity which can vary by several orders of magnitude. Due to the unusually low E/N value in the Vocus fIMR used in these experiments, absolute contributions of certain ions is challenging to ascertain due to the formation of both protonated molecules and  hydrate cluster ions of the same molecule alongside fragmentation. Unpicking these is not trivial, though the impact of hydrate formation upon the overall product distribution is not significant on the analysis carried out herein.

Considering these factors, we did not correct for sensitivity and use raw ion signal for both I-CIMS and Vocus measurements.
Gas phase I-CIMS observations were used for cross calibration between I-CIMS and Vocus under low-NOx conditions(S2), showing a generally consistent calibration factor for ions containing 1-6 oxygen atoms, indicating that these results are not influenced by vastly different I-CIMS sensitivities. This is further confirmed by similar ion signal distributions observed between both the I-CIMS and Vocus. Comparison of I-CIMS and Vocus data (Figure S1) shows that 30-60 % of the I-CIMS signal is from ions with chemical formulas which are also observed in the Vocus.

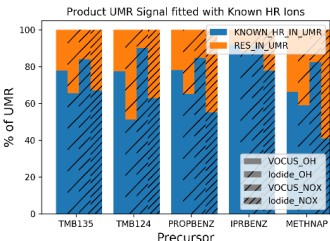


**Figure 1 Proportion of UMR signal fitted by identified HR ions for all precursors and experimental conditions from I-CIMS and Vocus**

The signal fitted by identified HR ions was compared with the total unit mass resolution (UMR) signal, excluding ions known to be unrelated to the experiment using OFR and instrument backgrounds. This comparison shows that between 40 and 90 %
of the observed signal has been assigned elemental formulae, with the residual signal (orange) of unidentified signal being either the result of poor signal to noise or complex overlapping peaks, making peak identification infeasible. This signal may also have some contribution from ion fragments in the Vocus and from thermal decomposition of desorbed products for the I-CIMS data. Analysis herein has been carried out for the blue proportion of signal, for which elemental formulae have been selected.





**3    Results**

**3.1    Overview of gas and particle phase oxidation products**

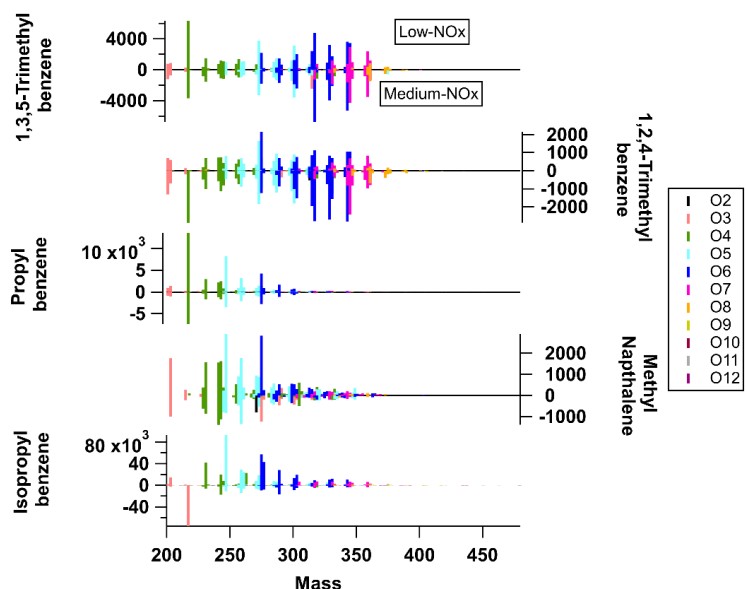

**Figure 2 Average mass spectra of particle phase measurements from I-CIMS for all precursors under low-NOx conditions (upwards) and medium-NOx conditions (downwards – negative values correspond to magnitude of positive signal)**

The gas phase (low-$NO_x$) mass spectra are dominated by small non-HOM oxidation products, with most of the signal in the $C_2$-$C_4$ range. A similar distribution is seen for the gas phase (medium-$NO_x$) mass spectra, although 1,3,5-TMB and *n*-propylbenzene show an increase in the contribution of the signal at higher C numbers that will be discussed below. As expected, the particle phase mass spectra show a higher fraction of signal at higher C number and an increase in the relative proportion of HOM signal. There are marked differences between the distributions of particle phase products, although there are many

common ion formulas observed, which will be discussed in section 4.1.2. The particle phase mass spectra obtained for each precursor are also included in Figure 2, with the particle phase (low-$NO_x$) spectra shown pointing upwards and the particle phase (medium-$NO_x$) spectra pointing downwards. The ion peaks have been coloured according to the number of O atoms found in the molecular formula. The two TMB isomers have very similar mass spectra. These two highly substituted ring species have a larger proportion of very oxidised ions ($O_6$-$O_8$) than the other precursors whose spectra are dominated by

compounds with $O_3$-$O_5$. N-containing ions do not contribute significantly to product signal under medium-$NO_x$ conditions (< 10 % except for 1-methyl naphthalene).

To compare the carbon skeleton of the oxidation products, the ion intensity of all ions with the same carbon number in the average mass spectra have been summed and are presented for each precursor in figure 3 for the gas phase and particle phases. This data has been further split into non-HOMS ($O \leq 5$) and HOMS ($O \geq 6$) based on the definition in Bianchi et al., 2019.

Comparison shows clear differences in distribution of products from different precursors, and in particular, between isomers of the $C_9$-aromatics.

Through comparison of these precursors under similar oxidation conditions, we can see that lumping of VOC isomers may not be valid in the context of SOA formation. It is possible that lumping VOCs based on single and multi- substituent could prove useful for model parameterisations. Furthermore, the evident HOM formation under elevated NOx conditions is important to

consider for effective policy implementation. Detailed trends in gas and particle composition for each precursor will be discussed in the following sections.



Atmospheric Chemistry and Physics Discussions — EGU Open Access

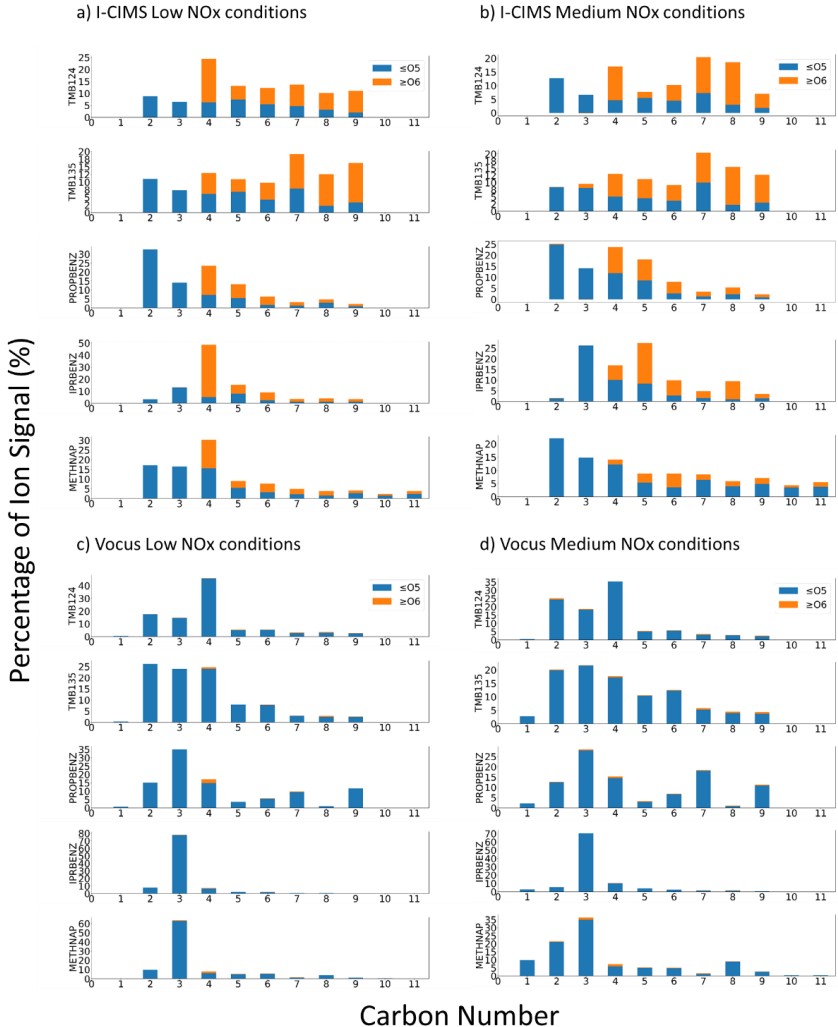

**Figure 3** Signal distribution across carbon number coloured by proportion of HOM signal for a) I-CIMS particle phase under low-NOx conditions b) I-CIMS particle phase under medium-NOx conditions c) Vocus gas phase under low-NOx conditions and d) Vocus gas phase under medium-NOx conditions [only signal contributions up to precursor carbon number are included (C9 for TMB124,TMB135,PROPBENZ & IPRBENZ, C11 for METHNAP) capturing both the ring-retaining and ring scission products for mechanistic discussions below. There is very little signal above C9 for the C9-aromatic precursors.]

### 3.2    Individual VOC SOA Product Composition

#### 3.2.1    1,2,4-trimethyl benzene

In the gas phase (low-$NO_x$) mass spectrum (Fig 3a), over 40 % of signal comes from $C_4$ ions, with the two largest ions being $C_4H_6O_2$ and $C_4H_8O_3$. In the particle phase (low-$NO_x$) MS, the signal is distributed more broadly across ions of different carbon numbers.  The largest contribution is also from the $C_4$ ions (25 %) with the $C_5$-$C_9$ all contributing ~ 15 % each. The majority of the $C_4$ product signal comes from $C_4H_4O_6$, with the other largest single ions being $C_5H_6O_5$, $C_7H_{10}O_6$ and $C_9H_{12}O_6$.

Under medium-$NO_x$ conditions, the dominant gas phase ions are consistent with those produced under low-$NO_x$ conditions and the product signal shows a broadly similar distribution to the medium-$NO_x$ conditions, with a small reduction in the fraction of $C_4$ product ions and an increase in $C_2$ product signal. A small contribution to signal comes from N-containing ions (3.8 %).





In the particle phase, the mass spectrum and C number distribution is fairly similar to the low-$NO_x$ conditions, although there is a doubling in the fraction of HOMs with $C_7$-$C_8$. In addition, 9.3 % of the particle product signal is from N-containing ions; specific ions have not been included in further analysis due to potential contributions from thermal decomposition.

### 3.2.2 1,3,5-trimethyl benzene

The most abundant contribution to the gas phase (low-$NO_x$) MS comes from an almost equal split between $C_2$-$C_4$ ions (together ~ 75 % of product signal), with the most dominant ions being $C_4H_6O_2$ and $C_3H_6O_2$. The spread of signal in particle phase (low-$NO_x$) MS is across ions with a broader range of carbon numbers, with $C_7$-$C_9$ HOM ions contributing ~ 40 % of product signal. The most dominant of these ions are $C_7H_{10}O_6$ and $C_9H_{12}O_6$.

In the gas phase medium-$NO_x$ mass spectrum, there is a shift with an increased contribution from ions with larger carbon numbers than under the low-$NO_x$ conditions. The most dominant gas phase ions were $C_4H_6O_2$, $C_3H_6O_2$, $C_2H_4O_3$ and $C_5H_6O_3$. Nitrogen containing ions contribute only 2.8 % of observed signal. In the particle phase (medium-$NO_x$) mass spectrum the most abundant ions are consistent with those under low-$NO_x$ conditions. There is a 7.8 % contribution to the product signal from nitrogen containing ions.

### 3.2.3 Propyl benzene

Similar to the TMB isomers, the majority of gas phase (low-$NO_x$) mass spectrum signal comes from $C_2$-$C_4$ ions, but in this case the $C_3$ ions have the highest intensity ($C_3H_8O_2$ and $C_3H_6O_2$). 10 % of signal contribution is from $C_9$ ions, with the most dominant ion being $C_9H_{10}O$, most likely propiophenone, the first generation ketone from OH attack on the $n$-propyl chain(Bloss et al., 2005). In contrast, the bulk of signal in the particle phase (low-$NO_x$) mass spectrum exists at $<C_5$, with a smaller contribution from HOM ions than the TMB isomers. The majority of ions observed correspond to predicted oxidation products in the MCM. Abundant oxidised ions not included in the MCM include $C_5H_6O_6$ and $C_5H_6O_5$.

The gas phase (medium-$NO_x$) mass spectrum product distribution is largely similar to low-$NO_x$ conditions, however there is an enhanced contribution from $C_7H_6O$, most likely benzaldehyde. N-containing ions contribute 3.5 % to gas phase product signal. In the particle phase (medium-$NO_x$) mass spectrum, the dominant ions are largely consistent with those observed under low-$NO_x$ conditions. N-containing ions contribute 5.8 % of the signal.

### 3.2.4 Isopropyl benzene

In the gas phase (low-$NO_x$) mass spectrum ~ 80 % of the product signal can be attributed to $C_3$ ions potentially formed from scission of the iso-propyl group from the ring, with the two largest ions being $C_3H_8O_2$ and $C_3H_6O_2$. In the particle phase (low-$NO_x$) mass spectrum, however, by far the largest contribution (~ 50 %) was from highly oxygenated $C_4$ ions, $C_4H_4O_6$ and $C_4H_6O_6$.

The gas phase (medium-$NO_x$) mass spectral product distribution was largely similar as under low-$NO_x$ conditions, with a slightly increased contribution from higher carbon number ions. There is a 3.6 % contribution to signal from N-containing ions. The particle phase (medium-$NO_x$) mass spectral product distribution was strikingly different compared to low-$NO_x$ conditions, with a larger contribution from $C_3$, $C_5$ and higher carbon number ions and the largest single ions were $C_3H_6O_3$, $C_4H_4O_4$ and $C_5H_6O_6$. In particular, there is a significant contribution from $C_5$-$C_8$ HOMs ions. Overall, there was a 2.5 % contribution to signal from N-containing ions.

### 3.2.5 Methyl Naphthalene

In the gas phase (low-$NO_x$) mass spectrum, ~ 60 % of gas phase product signal is attributable to just a few $C_3$ ions ($C_3H_8O_2$, $C_3H_6O_2$ and $C_3H_4O_3$). The particle phase shows a broader spread of product signal across the carbon number range, with a significant contribution of oxidised ($>O_6$) ions ranging from $C_4$-$C_{10}$, with the largest contribution being from $C_4H_4O_6$ and $C_4H_4O_5$.

The gas phase (medium-$NO_x$) mass spectrum showed an increased contribution from $C_2$ and $C_8$ ions compared with low-$NO_x$ conditions. The most dominant ions observed were largely consistent with low-$NO_x$ conditions. Overall, N-containing ions contributed a much larger fraction of the signal in both gas (16.2 %) and particle phases (22.3 %) than for the other precursors. In the particle phase (medium-$NO_x$) mass spectrum, the product distribution was again similar to low-$NO_x$ conditions, with a slight shift in signal distribution to higher carbon numbers. The most dominant ions in the particle phase were consistent with those observed under low-$NO_x$ conditions.





### 3.3 Comparison with Orbitrap Analysis

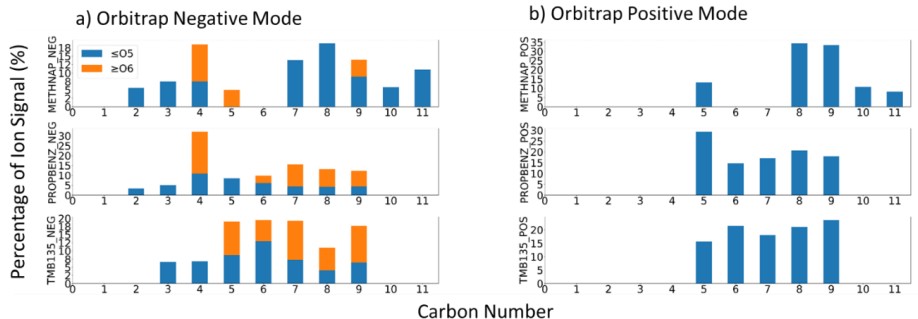

**Figure 4 Signal distribution across carbon number coloured by proportion of HOM signal for a) Orbitrap Negative mode and b) Orbitrap Positive Mode [only signal contributions up to precursor carbon number are included ($C_9$ for TMB135 and PROPBENZ, $C_{11}$ for METHNAP) capturing relevant products for mechanistic discussions below. Ions which contributed less than 0.5 % of observed signal were assumed to be background and removed.]**

Offline filters of SOA generated from the low-$NO_x$ oxidation of 1-methyl naphthalene, propyl benzene and 1,3,5-trimethyl benzene were analysed by Orbitrap LC-MS. The data presented shows all ions above 0.5 % of total ion signal; this was selected as a threshold to remove contributions from ions which may be contaminants. Though some ions are observed at higher carbon and oxygen numbers in Orbitrap measurements, these are not included in this analysis and we focus on products up to precursor carbon number as we can apply current mechanistic understanding in interpretation of these.

Results from both negative and positive modes are shown in Figure 4; negative mode shows most similarity to the composition observed by I-CIMS while positive mode looks very different. Positive mode is generally more sensitive to peroxide and carbonyl species, and similar ion distributions are observed for the $C_9$-aromatics, with a relatively larger contribution from $C_5$ ions in the case of propyl benzene. 1-methyl naphthalene shows a major contribution from $C_8$-$C_9$ products which are minor components in I-CIMS.

The distribution of ions for 1,3,5-trimethyl benzene is similar between I-CIMS and negative mode, with a larger relative contribution of $C_5$-$C_6$ ions and a smaller relative contribution of $C_2$-$C_3$ ions in negative mode compared with I-CIMS. Propyl benzene also shows very similar ion distributions to I-CIMS data, with a smaller contribution from $C_2$-$C_3$ ions and a larger relative $C_7$-$C_9$ contribution in both HOM and non-HOM ions. 1-methyl naphthalene shows significant differences between the I-CIMS and negative mode data in for higher carbon numbers ($C_7$ – $C_9$ ion contributions are small in I-CIMS but contribute significantly to negative mode) but are largely similar for low carbon number with $C_4$ ions contributing the dominant proportion of signal.

## 4 Discussion

### 4.1 Composition

#### 4.1.1 Comparison with the MCM

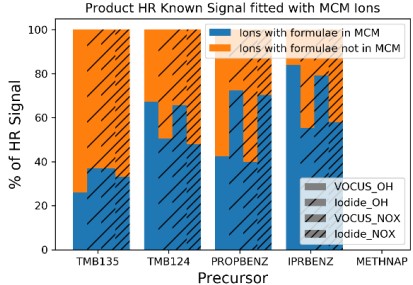

**Figure 5 Comparison of signal contribution of ions predicted to form by the MCM and those observed**





The Master Chemical Mechanism (MCM, mcm.york.ac.uk) is a near-explicit chemical mechanism which describes the detailed gas-phase chemical processes involved in the atmospheric degradation of a series of primary emitted VOCs. These include a large number of major emitted anthropogenic and biogenic species, and all of the $C_9$-aromatic isomers studied in these experiments.

The construction protocol developed to allow the building of comprehensive, consistent gas phase degradation schemes for aromatic VOCs in the MCM is given in Jenkin et al., 2003, which was subsequently updated, and evaluated/optimised using an extensive range of chamber experiments in 2005 by Bloss et al., (2005a); Bloss et al., (2005b). The general philosophy behind the MCM is to directly use the most up to date available literature information on the kinetics and products of elementary reactions relevant to VOC oxidation in order to build near-explicit representations of atmospheric degradation mechanisms. A fundamental assumption in the mechanism construction process is that the kinetics and products of a large number of unstudied reactions can be defined on the basis of the studied reactions of a smaller subset of similar chemical species, by analogy and with the use of structure-reactivity correlations (structure activity relationships (SARs)(Vereecken et al., 2018) to estimate the otherwise unknown parameters needed to construct the mechanisms.

Comparisons of the observed ions against the gas-phase oxidation products of aromatic hydrocarbons produced in the aromatic chemical schemes in the MCM can act as benchmarks to determine the relative importance of new pathways for different aromatic precursors. It also provides a basis for understanding where gas phase mechanisms could be improved in order to describe SOA formation. For the four average spectra of each precursor (1-methyl naphthalene is not within the MCM), the percentage of the mass spectral signal associated ion formulas also predicted by the MCM was calculated and is shown in Figure 5.

The proportion of mass spectral signal associated with ions consistent with MCM products was variable between precursors and techniques, ranging from 25 % (gas phase (low-$NO_x$) for 1,3,5-TMB) to 80 % (gas phase (low-$NO_x$) for isopropyl-benzene). The mechanisms contained within the MCM do not capture the same product profile for the trimethyl benzene isomers, though it should be noted that this agreement would not be expected in the particle phase as the MCM is a gas phase mechanism. Previous chamber experiments have shown good agreement between gas-phase measurements and MCM results, suggesting that its gas phase oxidation is well represented(Metzger et al., 2008; Rickard et al., 2010; Wyche et al., 2009). It may be that conditions in the OFR are dissimilar to these previous chamber experiments or that the measurements herein are more sensitive to previously undetected species. These are signal comparisons, and though in I-CIMS these may be effected by differences in sensitivity, the fact that the Vocus and I-CIMS see similar trends suggests that this does not affect our conclusions that the common ions contribute the majority of product signal as discussed in section 2.3.

### 4.1.2 Comparison between precursors

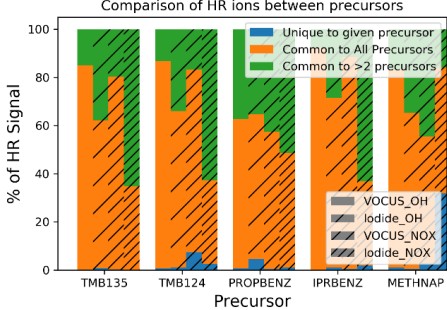

**Figure 6 Comparison of signal contributions of ions which were commonly observed from all precursors (orange), ions common to two or more precursors (green) and ions unique to the given precursor (blue) for all precursors and experiments**

Here we investigate the similarities and differences in the observed ions across precursors. Products within the MCM are expected to be common between the different aromatic precursors as the decomposition pathways built into the mechanism for the aromatics follow a specific mechanism building protocol (Bloss et al., 2005). The majority of product signal (average of 94 % for I-CIMS, average of 98 % for Vocus) comes from ions with molecular formula that are common between either multiple (green) or all (orange) of the precursors investigated, leaving an incredibly small contribution from ions which are unique (blue) to any given precursor. As discussed previously, most commonalities are between the $C_9$ precursors (Fig S4) and within this subset, between the trimethyl benzenes. In all cases, unique oxidation products from these different precursors contribute only a tiny amount to overall product signal even in cases where they may contribute significantly to the number of ions (S3). These patterns between observed products are consistent under both low and medium-$NO_x$ conditions, for the gas





and the particle phase. The majority of the common products are ring scission products while a large proportion are also HOMs.

HOMs have varying contribution to the SOA of different aromatic precursors; the proportions of the particle phase (low-$NO_x$) and (medium-$NO_x$) mass spectra that can be attributed to HOM are shown in Table 2. The fraction of the particle mass spectra assigned as HOMs is within a few percentage difference under the low and medium-$NO_x$ conditions for all precursors, except 1-methyl-naphthalene where the proportion drops by half when $NO_x$ is added (27 % versus 14%). The precursor with the highest fraction of HOMs is 1,2,4-TMB (34-36 %) and the lowest fraction is for propyl benzene (12-15%).


| Precursor | | TMB135 | TMB124 | PROPBENZ | IPRBENZ | METHNAP |
|---|---|---|---|---|---|---|
| % of Particle Phase Signal from HOM | Low-$NO_x$ Conditions | 28 | 34 | 12 | 28 | 27 |
| | Medium-$NO_x$ Conditions | 24 | 36 | 15 | 24 | 14 |

**Table 2 Percentage of particle phase I-CIMS signal described by HOMs (≥ O6 and not in the MCM)**





### 4.2 Mechanisms

Aromatic oxidation can proceed via hydrogen abstraction or OH-addition to the ring, with OH addition giving the highest branching ratio (Bloss et al., 2005). Upon OH addition, further oxidation proceeds via the phenolic, epoxy-oxy or bicyclic peroxy radical (BPR) pathways, which have varying relative yields for the different aromatics (Table 3). These different pathways can result in a variety of ring-retained or ring-scission products, both of which are expected to contribute to SOA (Calvert, 2002; Schwantes et al., 2017).

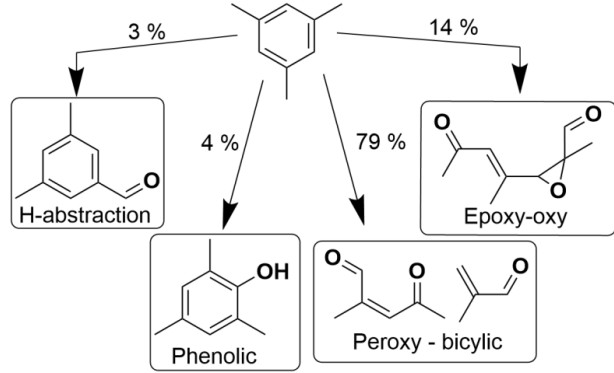


**Figure 7 Schematic of OH-initiated oxidaion pathways of 1,3,5-trimethyl benzene as included in MCM v3.1 adapted from**(Metzger et al., 2008)

| Precursor | H-Abstraction | Phenolic | Peroxy-bicyclic | Epoxy-oxy |
|---|---|---|---|---|
| TMB124 | 0.06 | 0.03 | 0.61 | 0.3 |
| TMB135 | 0.03 | 0.04 | 0.79 | 0.14 |
| PROPBENZ / IPRBENZ | 0.07 | 0.18 | 0.65 | 0.1 |

**Table 3 Relative branching ratios assigned to OH-initiated oxidation routes to first generation products in MCMv3.1(Bloss et al., 2005)**

Benzaldehyde and phenolic channels are predicted in the MCM to be less important for more substituted aromatics. Recent work has identified a potential source of ring-retained products from pathways of ipso-addition followed by dealkylation (Loison et al., 2012; Noda et al., 2009), however this has been reported as negligible and thus is not included in the new MCM SARs(Jenkin et al., 2018) . Yields from the epoxy-oxy pathway have been measured as significantly smaller than predicted by the MCM(Zaytsev et al., 2019) which are consistent with theoretical work(Li and Wang, 2014; Wu et al., 2014). The relative
yields of different pathways are not known for the propyl benzene isomers and are estimated based on toluene (Bloss et al., 2005). The epoxy-oxy radical route is included in the MCM to represent the balance of chemistry not accounted for by other routes(Jenkin et al., 2003). This highlights the importance of more detailed mechanistic work to evaluate the relative importance of different pathways for different isomers.

Decomposition via the BPR intermediate or its stabilised isomer is the dominant pathway for all precursors(Jenkin et al., 2018),
and recent work has shown that this pathway, which has previously been expected to yield only ring-scission products, can also contribute to the formation of ring-retaining HOMs via oxygen addition(Jenkin et al., 2018) and subsequent autoxidation(Molteni et al., 2018; Wang et al., 2017). It has been suggested that H-migration in BPRs of substituted aromatics may be faster and compete with HO$_2$/NO reactions, thereby resulting in increased HOM formation (Wang et al., 2017). Due to the formation of both ring-retaining and ring scission products from the BPR intermediate, it is challenging to predict which
precursors would be expected to form more ring-scission or ring retained products.

Ring scission products, mainly formed from the decomposition of BPR have been commonly observed from different aromatic precursors (Arey et al., 2009; Wang et al., 2007; Yu et al., 1997). Theoretical studies expect the decomposition of these intermediates into 1,2-dicarbonyls and co-products (Li and Wang, 2014; Wu et al., 2014). The larger co-products have been found at systematically lower yields than the corresponding 1,2-dicarbonyl products(Arey et al., 2009), suggesting further



photochemical processing is taking place. Though most studies of these dicarbonyls have been in the gas phase, they have recently been observed in the particle phase from oxidation of aromatics (Zaytsev et al., 2019).

In this study, we provide a comparison of composition for the aromatic precursors and compare the relative importance of different oxidation pathways for different $C_9$ aromatics. This comparison is important for understanding differences in SOA formation and composition from these often grouped precursors. A follow-up study will provide detailed insight into the

mechanisms responsible for formation of the dominant oxidised products from these precursors.

In order to evaluate the ions in a mechanistic context, they have been classified as ring scission or ring-retaining. Ring scission products can be unambiguously defined as ions $\leq C_5$. This leaves a subset of larger carbon number ions which could be ring-retaining of ring scission products. To obtain a more accurate representation of ring-retaining products, the double bond equivalency (DBE) was calculated, with ring-retaining ions having a DBE of at least 4. Further classification was attempted

through use of an Aromaticity Index (AI), however this proved unsuitable for HOMs which have high oxygen content. The discussion herein is based on ring scission products having $\leq C_5$ and ring-retaining products being $\geq C_6$ with a DBE of 4. However, there may be some species, for instance, from the epoxy-oxy route that will be mislabelled using this approach, but their contribution is expected to be minor.

### 4.2.1    Ring Scission Products

Ring-scission from different aromatics is known to yield a range of alpha-dicarbonyl products and alpha, beta-unsaturated-gamma dicarbonyl, furanone and small acid co-products (Arey et al., 2009; Calvert, 2002; Smith et al., 1998, 1999), many of which are included within the MCM(Bloss et al., 2005). We compare our data to scission products common to a range of aromatics identified by Arey at al. 2009 and all of these were observed in our experiments by I-CIMS (I), Vocus (V) or both (V+I) (Table 4). Ring-scission products are largely dominant in the gas phase, though they play a role in the particle phase

composition, particularly so for the less substituted aromatics. In some cases, such as 1,3,5-TMB, known ring scission ions given in Arey et al. are abundant products, for example $C_5H_6O_3$ is the 10[th] largest signal in the particle phase under low-NO$_x$ conditions. In the MCM, BPR is near exclusively decomposed into ring-scission products which proves suitable for describing the major observed particle phase products for propyl and isopropyl benzene ($C_5H_6O_6$, $C_5H_6O_5$ and $C_4H_4O_6$) and explains the larger proportion of product signal observed (Figure 5).

A large proportion of the ring scission products observed in the particle phase are more oxidised than those previously reported(Arey et al., 2009) and many fit the definition of HOM by Bianchi et al., 2019. These products appear to have undergone oxygen addition to the same carbon backbone as many of the small gas phase dicarbonyls, suggesting that further gas phase oxidation and partitioning of dicarbonyl co-products may be an important contributor to SOA. Further processing of these ring-opened products may explain the discrepancy between 1,2-dicarbonyls and their co-products observed in many

gas phase studies of aromatic of aromatic oxidation(Arey et al., 2009). This has been shown previously for SOA formed from 1,3,5-trimethyl benzene where 3-methyl maleic anhydride ($C_5H_4O_3$) , 2-methyl-4-oxo-2-pentenal ($C_6H_8O_2$) and 3,5-dimethyl-5(2H)-2-furanone ($C_6H_8O_2$) contribute to SOA formation and growth through further oxidative processing(Johnson et al., 2005; Rickard et al., 2010). Further examples of this are exhibited in our results, including $C_7H_{10}O_2$, which is observed from both trimethyl benzene isomers in Vocus gas phase mass spectra, and $C_7H_{10}O_{(4-7)}$ formulas that are some of the most dominant

product ions observed in SOA from both trimethyl benzene isomers. In the case of 1,2,4-TMB, $C_5H_6O_2$ is observed in the gas phase while $C_5H_6O_{(3-7)}$ are dominant ions in the particle phase. This is also the case for propyl benzene, where $C_5H_6O_5$ and $C_5H_6O_6$ are the two largest contributors to the particle phase. Other observed products can be explained by further oxidation of scission products within the MCM, such as $C_4H_6O_6$ which is a major ion from isopropyl benzene oxidation.

| Dicarbonyl Products | TMB124 | TMB135 | PRBZ | MN | IPRBZ |
|---|---|---|---|---|---|
| $C_5H_6O_2$ | V | V | V | V | V |
| $C_6H_8O_2$ | V | V+I | V | V | V |
| $C_7H_{10}O_2$ | V+I | V | V+I | V | V+I |
| $C_5H_6O_3$ | V +I | V+I | V+I | V | V |
| $C_6H_8O_3$ | V+I | V+I | V+I | V+I | V+I |





**Table 4 Dicarbonyl products observed in gas and particle phase from Arey (2009) under low-NO$_x$ conditions (V and I stand for Vocus and I-CIMS respectively)**

The most dominant of the scission products we observe in gas and particle phases are lower carbon number than those within the MCM, with C$_3$ products being most dominant in the gas phase for PROPBENZ, IPRBENZ and METHNAP, and C$_4$ products being more important in the TMBs. In the particle phase, C$_4$ scission products have the largest contribution for
TMB124 and METHNAP. Scission products we observe have previously been detected from oxidation of 1,2,4-trimethyl benzene including C$_5$H$_8$O$_3$, C$_4$H$_6$O$_3$ and C$_3$H$_4$O$_2$ (Zaytsev et al., 2019). Ring scission remains a dominant pathway under the NOx conditions, though the spread across carbon numbers increases in the gas and particle phases as shown in the carbon number distributions. Ring-scission HOMs are most commonly observed in the particle phase for the C$_4$ and C$_5$ products and are particularly important in the case of the C$_9$ aromatics. Newly proposed epoxy-dicarbonyl products (Li and Wang, 2014)
have also been observed in this study (C$_4$H$_4$O$_3$ – epoxybutanedial, C$_5$H$_6$O$_3$ – methyl epoxybutanedial and C$_6$H$_8$O$_3$ –dimethyl epoxybutanedial) from all aromatic precursors in the particle phase.

### 4.2.2 Ring-retaining Products

Ring-retaining products can form from two different pathways: hydrogen abstraction (from the attached alkyl groups) which can result in fragmented ring-retaining products or via the BPR pathway which results in non-fragmented ring-retaining
products.

Ring-retaining HOMs from 1,3,5-trimethyl benzene oxidation have been identified from experiments carried out in a range of different chambers and conditions (Hammes et al., 2019; Molteni et al., 2018; Sato et al., 2012). Our observations show good agreement with ion formulas observed in these experiments, with divergence at significantly higher NO$_x$ conditions (Hammes et al., 2019). This includes observations of ions such as C$_9$H$_{12}$O$_{(2-8)}$ (Molteni et al., 2018) alongside ions observed by LC-ToF-
MS(Sato et al., 2012, 2019) which were all important in our experiments (Table S1).Ring retention was previously found to contribute ~ 25 % of SOA mass from a study 1,2,4-trimethyl benzene under elevated NO$_x$ conditions which were attributed to BPR, phenolic and benzaldehyde channels despite low ring-retaining concentrations in the gas phase, consistent with our observations. These results identified ions such as C$_9$H$_{12}$O$_{(4-6)}$ which are also dominant in our observations for both 1,2,4- and 1,3,5-trimethyl benzenes, though we also observed a significant contribution from more oxidised ions such as C$_9$H$_{12}$O$_6$ and
C$_9$H$_{14}$O$_6$ (Zaytsev et al., 2019).

Previous results have suggested that C$_8$ and C$_7$ compounds such as C$_8$H$_{10}$O$_{(4-5)}$ and C$_7$H$_8$O$_{(4-5)}$ from 1,2,4-trimethyl benzene oxidation can be formed from the dealkylation pathway, though they may also be formed from multiple pathways (Noda et al., 2009; Zaytsev et al., 2019)). However, other studies have shown that dealkylation is not a significant pathway (Aschmann et al., 2010; Loison et al., 2012). In our results, C$_8$ ions are only important for the trimethyl benzenes indicating that this
pathway may be important for these precursors but not for others. Though we observe the less oxidised C$_8$ products, the major C$_{7-8}$ products we observe are C$_8$H$_{10}$O$_6$ for TMB124 and C$_8$H$_{10}$O$_7$ for TMB135.

To gain a broader overview of the importance of ring-retaining products across the various precursors, we have estimated the contribution of ring-retaining ions to total product signal. Our results show that ring retained HOM formation is most important for TMB135 where these ions contribute 23 % of total observed product signal in the particle phase measured by I-CIMS, with
a smaller contribution of 13 % to TMB124, and a minor contribution to IPRBENZ (13 %) and PROPBENZ (11 %) under low-NO$_x$ conditions.

This indicates that ring-retaining product formation is important for the aromatics hydrocarbons with more substituent groups on the ring rather than a single alkyl side chain, suggesting that the substitution plays a role in favouring retention of the ring during both H-abstraction and OH addition. This may be due to stabilisation of substituent ring or due to the steric hindrance
of the methyl groups which make OH attack less favourable and result in a larger than estimated branching ratio of H-abstraction from the methyl group. It is apparent in our results that ring-retained HOMs persist upon perturbation by NO, suggesting that favourable transition state geometry of the more substituted aromatics may play a role in ensuring autoxidation of BPR remains competitive at higher NO conditions. This is not observed for the less substituted aromatics, which generally produce less HOM in the presence of NO$_x$.

Under low-NO$_x$ conditions, the relative importance of autoxidation for formation of ring-retaining HOMs from BPR intermediates varies for the different precursors and such impacts upon the relative yields of ring-retaining vs. ring-scission products. In the cases of propyl and isopropyl benzene, ring-retained HOM formation has a minor contribution to products highlighting the role that substituent groups on the aromatic ring play in favouring autoxidation leading to ring-retained HOM formation. Isopropyl benzene SOA has a larger contribution from ring-opened HOMs in the particle phase despite a larger
contribution from ring-retaining products in the gas phase, suggesting though their formation occurs, they undergo further oxidation and scission before partitioning.



Though we lack knowledge of specific pathways for methyl naphthalene, our observations of products which are assumed to contain a single ring ($\leq$ C10) are consistent with that of the $C_9$-aromatics, suggesting that beyond the opening of the first aromatic ring, the oxidation proceeds via similar pathways to that of the $C_9$-aromatics. Overlap in observations between $C_9$-
aromatics and the PAH could prove useful as a basic description of the major products formed in the photo oxidation of methyl naphthalene.

### 4.3 Implications for Ambient Observations

Aromatic oxidation leads to formation of oxidised products, which are largely common to all aromatic precursors. This makes identifying their contribution to ambient SOA challenging using online techniques. Furthermore, unique product ions
contribute a tiny proportion of signal and thus it is challenging to differentiate between different aromatics. However, markers are widely used for the interpretation of ambient measurements, and many of the ions we observe correspond to formulas which have been reported as markers of other precursors or oxidation pathways. This is shown in Table 5 where we compare the major ions observed from all aromatics with previous literature and suggests that some of these source attributions may not be valid when aromatics are present.

A rapid increase in the development and application of novel mass spectrometric techniques has given a wealth of near-molecular level detail which is unprecedented in atmospheric chemistry (Isaacman-VanWertz et al., 2017; Laskin et al., 2018). Such high resolution observational datasets are extremely useful for the development and evaluation of detailed chemical mechanisms used to understand and simulate gas and aerosol phase composition in a wide variety of science and policy applications related to air quality and climate. Comparing our results with other laboratory and field experimental observations
shows that some major ions from aromatic oxidation are concurrent with other sources or VOC systems which has implications for marker identification in ambient datasets. For example, $C_9H_{12}O_6$, observed in all aromatic oxidation experiments which has previously been identified in cases of biomass burning(Kourtchev et al., 2016) and also as a product of limonene oxidation(Hamilton et al., 2011). $C_7H_8O_6$ and $C_9H_{12}O_7$ have been observed from aromatic autoxidation (Molteni et al., 2018), however $C_7H_8O_6$ is also as a product of alpha phellandrene ozonolysis (Mackenzie-Rae et al., 2018). Another product observed
from all aromatic precursors is $C_8H_{12}O_6$ which is widely reported as MBTCA, a marker of monoterpene SOA (Hu et al., 2013; Kourtchev et al., 2016). The majority of these observations are from CIMS approaches which lack the separation of LC or GC.

Though we observe some unique ion signal, this signal is distributed amongst many ions so no single ion has a large enough signal intensity to be used as a marker, and these low signal intensity ions are thus likely to be absent in complex ambient spectra. Even in single component experiments they are not ideal unambiguous markers of SOA, thus, in more complex and
ambient SOA samples, it is hard to see how they could be used effectively. Furthermore, due to these ions being minor components, they are difficult to assign formulas due to a relatively poor SNR and as many of them are small carbon numbers, they could be the result of low signal intensity. Thus, comparison of precursors suggests that identifying tracers from aromatic precursors using online techniques with no pre-separation is not possible. Most striking is the fact that methyl naphthalene, which is a $C_{11}$ molecule has > 68 % of signal common with the other precursors in all cases, despite the fact that its oxidation
will proceed differently. Tracer based detection of SOA is thus not likely to be useful in ambient measurements, and though potentially isomeric information may provide unique markers, ion formulas themselves are unlikely to uniquely identify sources between different aromatic precursors.

| Ion Formula common to all aromatic experiments | Sources Previously Reported | References |
|---|---|---|
| $C_9H_{12}O_6$ | Limonene | (Hamilton et al., 2011) |
| $C_8H_{10}O_6$ | Isoprene | (Nguyen et al., 2011) |
| $C_8H_{10}O_7$ | Biomass burning, Guaiacol | (Qi et al., 2019; Romonosky et al., 2014) |
| $C_7H_8O_6$ | Guaiacol | (Romonosky et al., 2014) |
| $C_9H_{12}O_7$ | a-pinene | (Romonosky et al., 2014) |
| $C_4H_4O_6$ | Tartaric acid | (Cheng et al., 2016) |
| $C_8H_{12}O_6$ | a-pinene | (Szmigielski et al., 2007) |
| $C_3H_4O_4$ | Pinene, burning, seasalt | (Isaacman-Vanwertz et al., 2018; Legrand et al., 2007) |




| $C_6H_6O_6$ | Biomass burning | (Qi et al., 2019) |
|---|---|---|
| $C_7H_8O_7$ | Biomass burning | (Qi et al., 2019) |
| $C_7H_8O_5$ | o-cresol | (Schwantes et al., 2017) |
| $C_4H_4O_4$ | a-Pinene | (Zhang et al., 2015) |
| $C_4H_4O_5$ | a-pinene | (Takeuchi and Ng, 2019) |
| $C_3H_4O_5$ | Maleic acid | (Gallimore et al., 2011) |
| $C_5H_4O_4$ | Biomass burning | (Priestley et al., 2018) |
| $C_8H_{12}O_5$ | a-pinene | (Shilling et al., 2009) |
| $C_8H_{10}O_5$ | a/b-pinene | (Takeuchi and Ng, 2019) |
| $C_8H_8O_7$ | Biomass burning | (Qi et al., 2019) |
| $C_7H_8O_8$ | a-pinene | (Krechmer et al., 2016) |
| $C_4H_6O_6$ | Isoprene | (Krechmer et al., 2016) |
| $C_7H_{10}O_4$ | Limonene | (Faxon et al., 2018) |

**Table 5 Comparison of ions observed commonly from all aromatic precursors with these ions reported in literature**

## 5    Conclusions

Oxidation of different $C_9$-aromatic isomers results in formation of similar products, however the signal distribution of these products in the gas phase and particle phases are distinct. Gas and particle phase compositions are distinct from one another, with many HOMs being observed almost exclusively in the particle phase I-CIMS measurements and many small scission products exclusively in the Vocus measurements.

Comparison of observations with the gas phase MCM mechanism highlights that the MCM does not capture the full extent of
observed ions, particularly the more oxidised HOM products formed from autoxidation pathways which have not been included in this mechanism. Comparing the online I-CIMS and Offline Orbitrap UPLC-MS observations shows good agreement in negative mode, with a large overlap in ring-retaining HOM ions observed by both techniques.

Evaluating products in terms of their formation mechanisms shows that oxidation of the more substituted aromatic precursors leads to a larger contribution of ring-retained products, while the single substituent aromatics yield large proportions of scission
products. Overall, ring-retaining products are the minor contributor to SOA product signal (< 25 %) while known scission products appear to undergo further oxidation and are major contributors to SOA from all aromatic precursors.

Perturbation of the oxidation system with NOx shows that HOM formation from aromatics proceeds at higher NO and this effect is most important for the more substituted trimethyl benzene isomers which continue to form ring-retained HOMs at higher NOx. The variation in composition between the $C_9$-aromatic isomers suggests that grouping VOCs by carbon number
for SOA formation may not be valid and the number of substituent groups may be a more valid grouping to capture their different SOA product distributions. Comparison with 1-methyl naphthalene shows that the products from a different starting precursor are largely similar, with the distributions being the key to differentiation of SOA from different precursors.

Comparison of ions observed from aromatic oxidation with literature shows overlap with those from other sources, suggesting that some source attributions may not be valid when aromatics are present. Correspondences between laboratory and ambient
spectra have been observed for oxidation products of biogenic VOCs(Yan et al., 2016), however this approach has not been taken for anthropogenic VOCs. Through co-measurement of many ions, these results are expected to support studies evaluating the importance of aromatics for urban SOA formation. This study has focused on single components; however, future work is required to identify the impact that mixing aromatics with biogenics has upon SOA composition(McFiggans et al., 2019).



## 6    Author Contribution

AM and HC designed the experiments. Instrument deployment and operation were carried out by AM, YW, JEK, AL, FM, PC, MAM, and MRC. I-CIMS data analysis was carried out by AM and Vocus data analysis by YW. DJB prepared the aerosol samples for LC-MS analysis. KLP designed the non-targeted LC-MS analytical and data processing methodology, analysed the samples and provided the processed data. AM, YW and JFH wrote the paper. All co-authors discussed the results and comments on the manuscript. The authors declare that they have no conflicts of interest.

## 7    Acknowledgements

Archit Mehra is fully funded by the Natural Environment Research Council (NERC) and acknowledges his funding through the NERC EAO Doctoral Training Partnership (NE/L002469/1) and CASE partnership support from Aerodyne Research Inc. Lin Wang acknowledges the Newton Advanced Fellowship (NA140106). Hugh Coe and Archit Mehra acknowledge funding as part of AIRPRO (NE/N00695X/1).

## 8    Competing Interests

The authors declare that they have no conflict of interest.

## 9    Data Availability

Data is available upon request from the corresponding author.

## 10    Supplement

Supplement is attached.

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
