# Peer review of "Evaluation of the Chemical Composition of Gas and Particle Phase Products of Aromatic Oxidation"

_Atmospheric Chemistry and Physics, 2020_

## Referee Comment (RC1) · Anonymous Referee #1 · 10 Apr 2020

This paper reports results of mass spectral analyses of gas- and particle-phase products formed in the reactions of several C9 alkylbenzenes and 1-methyl naphthalene with OH radicals in the presence and absence of NOx. The products are identified by mass and the atomic numbers corresponding to the masses, but other structural information is not provided. A large number of products are observed, with carbon numbers ranging from 2 to the number of carbons in the starting compound, and although the product distributions differ depending on the compound and NOx levels, the contributions of products that are unique to any given compound are relatively small. The product distributions are discussed in terms of extent of oxidation, whether the product is likely to be from fragmentation or ring retaining (based on atom numbers), the ex-

tent to which the products are consistent with MCM, and how the product distributions compare with what is observed in the atmosphere.

This paper gives potentially useful information on products formed from aromatics, but I have some concerns about how representative the experiments are of atmospheric conditions and the correspondence between the ion signals and actual product yields. In addition, I think the presentation and discussion of the results could be improved, especially with regard to mechanistic implications. The major issues I see are discussed below, followed by a summary of other issues or suggestions.

Major comments

There should be more discussion of how their experimental system differs from the atmosphere, and also the extent of secondary reaction of products formed. Can the very low wavelength UV light they use to generate the radicals (and NOx) photolyze the reactants or products and cause products to be formed that would not formed in the lower atmosphere or deplete products that otherwise be important? They give an "OH exposure" number for their experiments and state that it is similar to "Chinese megacities", but they do not give the range of OH exposure numbers in Chinese cities or elsewhere or citations for them. What is the fraction of initially present aromatic hydrocarbon that reacts during an experiment? Do they have an estimate of how much of observed products are from multi-generations of reaction? Have they attempted to model their experimental conditions to obtain information about representativeness?

The major results are presented primarily as figures giving fractions of ion signals that have various characteristics, plus some summary information given in the text. The paper has a "Supplementary Data" (SD) section to give additional information, and ideally it should have the information that is summarized or shown graphically in the text, so the reader can examine it in more detail, to either to verify the discussion in the paper or perhaps to gain other insights. The SD does have 20 tables giving the "top 20" product distributions for each of the 2 types of experiments with 2 analysis

methods and 5 compounds, but it only has the information regarding the ion detected and true/false flags indicating whether it is common or unique among the compounds studied and whether it has the same molecular formula as a product predicted by MCM. That is not the most interesting information they could present. These tables should include at least the relative ion signal intensity, and ideally also the classifications as ring-scission, ring-retaining, HOM, DBE, and other classifications they discussed or summarized. Instead of just indicating that this may be predicted by MCM, they should give the name and structure of the products(s) corresponding to this molecular weight. This would make the tables much more interesting and greatly increase the value of this work and information content of the paper.

The discussion of MCM and mechanistic implications could be improved. It is not surprising that MCM does not predict the full range of products they observe, expecially HOM, because (1) the version of MCM that is currently online does not have autooxidation reactions that are now believed to be important, and (2) it employs lumping or reduction methods when it gets to 3+ generation products of the compounds represented. What might be more interesting would be high yield products that MCM predicts that they DO NOT observe. These should be listed, or it should be stated that there are no such products if that is in fact observed. One way to do this would be to run MCM to model the conditions of the experiments, summarize the yields predicted, and list these against the observed relative ion signals of products with the same molecular weight in the experiments. Are the products observed more consistent with the revised mechanisms predicted by Wang et al (2017)?

The tables in the SD indicate that no signals were observed for glyoxal (C2H2O2), which is known to be formed in significant yields from all products (and is predicted by MCM). Also, methyl glyoxal (C3H4O2) should also be seen as a product from the trimethylbenzenes and propyl glyoxal (C5H8O2) should be formed from propyl benzene. Does this method not work for alpha-dicarbonyls? If so, state this. Or are they all rapidly photolyzed away by the high UV in their system? This gives me some doubt

about the credibility of the experimental system.

It is stated that they did not correct for sensitivity of the ion signals to the actual concentrations in their analyses, apparently because of the complexities involved. However, since they are using the ion signal as a surrogate for the yields, it would be useful to estimate the uncertainty or potential for error when not making this correction.

Other Comments

The abstract should state what they mean by "product signal" and describe the analysis methods in a few words.

The statement in the abstract that the MCM "highlights" missing product pathways, but it is not clear from that statement if it is the MCM that is missing something (which is the case), or the experiments aren't finding something that MCM predicts (which may or may not be the case – see comments above – but is probably not what the authors meant to say).

The observation that "A large proportion of the ring scission products observed in the particle phase are more oxidised than those previously reported" is significant and should be included in the abstract.

Was there only a single experiment of each of the 10 types listed on Table 1, or are the concentrations given there the averages of several experiments. If the latter, indicate the number of experiments?

The legend on the top right on Figure 1 should be moved to where it doesn't get in the way of the data being shown.

The labels indicating the compounds on Figure 3 need to be larger and easier to read. Right now they are in a very tiny font hugging the axes and are hard to see.

Could they use a different color on Figure 3 to indicate N-containing ions?

Table 2 would have more impact if it were shown as a bar-plot figure. It clearly shows

n-propyl benzene as an outlier among the alkylbenzenes in terms of NOx effects. (I would have thought isopropyl benzene would be more likely the outlier, since the autooxidation proposed by Wang et al (2017) may be more important for this compound.) It might be useful to also include their estimates of ring-retaining and ring-scission fractions on this same table or figure, since there are also discussed in the text.

It is not clear what is the purpose of table 3, giving MCM branching ratios for various types of reactions, since they give no comparison of this with their data. If they can identify their products by the classifications on this table, why don't they give the relative amounts derived from their data? If not, why include this table and discussion?

The "Authors Contributions" section does not list all the authors.

The supplement document does not identify the title, authors, and journal of the manuscript it goes with.

---

## Referee Comment (RC2) · Anonymous Referee #2 · 16 Apr 2020

Archit Mehra and co-workers conducted the evaluation of the chemical composition of gas and particle phase products of aromatic oxidation. Gas and particle phase composition are compared with the simulation results by Master Chemical Mechanism (MCMv3.3.1). This work highlights a series of missing highly oxidized products in the pathways. The work is therefore valuable in this regard. The article can be published once the authors have addressed the following points.

1. Line 31: "HOMs" Abbreviations should be given their full names when they first appear. Also, for Line 47 " SAPRC ".

2. For the experiment of 1-methylnaphthalene: why the mixing ratio of VOCs is lower

than other experiments? And for each VOCs, the experiments were conducted only once? If not, in all the experiments the mixing ratio of VOCs was controlled uniformly?

3. Line 141ïijŽwhat is the "PFA"?

4. Line 144: you should change the "N2" into "N2", and you should check the paper to avoid the similar error.

5. Figure 2: What is the ordinate?

6. Figure 3, Figure 4: The quality of images needs to be improved.

7. Line 273: how to identify the C4H6O2 and C4H8O3? Give the spectrum information.

8. Line 277-278: "with a small reduction in the fraction of C4 product ions and an increase in C2 product signal" what is the reason?

9. In this paper, are there any new mechanism that could optimize the MCM? When the missing highly oxidized products were considered in MCM, is there any difference between simulation and experiments?

---

## Author Comment (AC1) · 18 Jun 2020

**Response to Reviewers of:**

**"Evaluation of the Chemical Composition of Gas and Particle Phase Products of Aromatic Oxidation" by Archit Mehra et al., 2020, submitted to ACP**

**General Response**

We thank the reviewers for their comments, helping us to clarify ambiguities and further improve the manuscript. Referee #1 states that the manuscript provides potentially useful information however they have some concerns about how representative the experimental conditions are. Referee #2 feels the work is valuable as it identifies a series of missing highly oxidised products from current mechanisms.

**Overall Content and Scope**

The manuscript aims to provide a detailed characterisation of product distributions from the oxidation of a range of aromatic VOCs. The manuscript did not aim to relate in any way the observed ion signal strengths to concentrations, yields or act as a surrogate for these. As such, the comparison with the master chemical mechanism was solely based upon potential products which could be formed, and the model simulation itself was not run as the purpose of this study was not to make any quantitative comparison with yields estimated by the MCM. We feel that both reviewers have slightly misunderstood this, which we recognise could be a reflection of the way in which the manuscript has been written. To address this, we have made changes to make the aims clearer and added explicit clarification that we do not link the ion signals observed directly to product yields.

**Anonymous Referee #1**

We thank the referee for their detailed and helpful comments that we have used to clarify and improve the manuscript. As is stated above the main comments have stemmed from misunderstanding that the paper was investigating yields and hence required us to be explicit about the relationship between ion signals and concentrations and also the actual experimental conditions associated with implementing an MCM model run. As we have stated above we were not seeking to do this in the paper and have clarified this in the paper explicitly to address future misinterpretation. We have carefully addressed the referee's well considered comments.

*This paper reports results of mass spectral analyses of gas- and particle-phase products formed in the reactions of several C9 alkylbenzenes and 1-methyl naphthalene with OH radicals in the presence and absence of NOx. The products are identified by mass and the atomic numbers corresponding to the masses, but other structural information is not provided. A large number of products are observed, with carbon numbers ranging from 2 to the number of carbons in the starting compound, and although the product distributions differ depending on the compound and NOx levels, the contributions of products that are unique to any given compound are relatively small. The product distributions are discussed in terms of extent of oxidation, whether the product is likely to be*

*from fragmentation or ring retaining (based on atom numbers), the extent to which the products are consistent with MCM, and how the product distributions compare with what is observed in the atmosphere.*

*This paper gives potentially useful information on products formed from aromatics, but I have some concerns about how representative the experiments are of atmospheric conditions and the correspondence between the ion signals and actual product yields.*

Comments relating to how representative the experiments are of atmospheric conditions have been addressed in detail below. The referee also comments on the correspondence between ion signals and actual product yields - as we have already stated above, this was not the aim of the study and we have added additional clarification of this within the manuscript.

*In addition, I think the presentation and discussion of the results could be improved, especially with regard to mechanistic implications. The major issues I see are discussed below, followed by a summary of other issues or suggestions.*

We have addressed these points in turn below.

**Major comments**

*There should be more discussion of how their experimental system differs from the atmosphere, and also the extent of secondary reaction of products formed. Can the very low wavelength UV light they use to generate the radicals (and NOx) photolyze the reactants or products and cause products to be formed that would not formed in the lower atmosphere or deplete products that otherwise be important?*

The potential of the low wavelength UV light used to generate radicals to photolyse the reactants has been determined through calculation of the potential non-OH fates of the VOCs in the experimental system. These have been determined for the VOCs for which photolysis data exists (1-methyl naphthalene, 1,3,5-trimethyl benzene and 1,2,4-trimethyl benzene). The photolysis yields from these species are minor under the conditions used in this study, with reaction with OH contributing 99.5 %, 98.3 %, 99.9 % of the fate of 1,3,5-trimethyl benzene, 1,2,4-trimethyl benzene and 1-methyl naphthalene, respectively. Sufficient literature data on the photolysis of propyl and isopropyl benzene is not available and thus these estimates cannot be produced. However, we anticipate that their loss, like that of the other substituted aromatics is also dominated by OH oxidation under the conditions operated in these experiments. On this basis we conclude that photolysis is not important for the reactions of aromatic VOC in this study. The referee also asks about the potential for non-representative product photolysis or non-OH depletion of products. As has been discussed at length in the recent extensive review by Peng & Jimenez, 2020 and in Peng et al. 2016, the effect of non-representative UV photolysis of products or intermediates is very difficult to fully characterise as quite simply there is scarcity of data available. We thus conclude that despite this shortcoming there is no substantial evidence for large mechanistic differences and OFRs provide a useful tool for studies of this type.

We have added the following into the methodology to clarify this:

"The use of non-tropospheric low wavelength UV light to generate radicals in OFR can result in the potential photolysis of reactants. However, operational conditions can be optimised to ensure OH loss is the dominant removal process of the VOC (Peng et al., 2016; Peng & Jimenez, 2020). In this study, under the conditions used > 98 % the loss of VOC can be attributed to reaction with OH. This is based on calculations under the relevant conditions for 1,3,5-trimethyl benzene, 1,2,4-trimethyl benzene and 1-methyl naphthalene for which photolysis rate data exist. The relevant photolysis data does not exist for propyl benzene and isopropyl benzene and thus, to the extent that results for the TMBs and methyl naphthalene are also applicable to propyl and isopropyl benzene, we anticipate that their loss is also dominated by OH oxidation. The photolysis of products or intermediates are more difficult to fully clarify, as discussed in Peng et al. (2020) and thus the practical approach taken here is to optimise the OH oxidation to ensure that OH loss is the dominant pathway for the VOCs themselves. "

*They give an "OH exposure" number for their experiments and state that it is similar to "Chinese megacities", but they do not give the range of OH exposure numbers in Chinese cities or elsewhere or citations for them.*

We have added a reference to (Lu et al., 2019) and inserted the following sentence:

"For example, OH concentrations in Beijing can reach as high as ~ $1 \times 10^7$ molecules cm$^{-3}$ (Bryant et al., 2019), an order of magnitude higher than the global average of ~ $1 \times 10^6$ molecules cm$^{-3}$ (Lelieveld, Gromov, Pozzer, & Taraborrelli, 2016) which is used to relate OH exposure in an OFR to days of equivalent atmospheric oxidation (Lambe et al., 2015). This changes what would be 11 days of equivalent atmospheric oxidation in the OFR under global average OH concentrations to just over 1 day of oxidation in Beijing."

*What is the fraction of initially present aromatic hydrocarbon that reacts during an experiment?*

As discussed above, this study was focused upon product distribution and not yields thus the fraction of aromatic hydrocarbon reacted has not been quantified, and is not required for interpretation of the results presented in this manuscript.

*Do they have an estimate of how much of observed products are from multi-generations of reaction?*

We have made a lower bound estimate of this and added the following to the paper:

"Molteni et al. (2018) suggested that ring-retaining products with ion formulas containing 4-6 more hydrogen atoms more than their parent hydrocarbon could be attributed to multiple-OH attacks. More recently, Tsiligiannis et al. (2019) has shown that $C_9H_{14}O_z$ products from the oxidation of 1,3,5-trimethyl benzene have characteristics of second generation products owing to their enhanced contributions in experiments with higher OH exposures. Thus, in order to obtain a lower bound estimate of the contribution multi-generational OH attack, ions with 4-6 more hydrogen atoms than the parent hydrocarbon were classed as multi-generational products. Overall their contribution to signal is < 10 % in all experiments (Figure S5– S6)."

Added to supplement:

[Figure]

Figure S5 – Signal contribution from ions associated with multiple generations of reaction with OH under low-NOx conditions, defined as ions which contain > 4 hydrogen atoms more than the VOC precursor species (14 in all cases)

[Figure]

Figure S6 – Signal contribution from ions associated with multiple generations of reaction with OH under med-NOx conditions, defined as ions which contain > 4 hydrogen atoms more than the VOC precursor species (14 in all cases)

*Have they attempted to model their experimental conditions to obtain information about representativeness?*

The objective in this work was not to look at yield comparisons under different conditions, but rather understand the range of product distributions comparing those observed experimentally with those defined in the current version of the MCM, therefore no modelling was performed. Comparison with the MCM was with that of all potential products output in the subset mechanisms, and thus not related to any given experimental condition, but to act as a benchmark of current scientific understanding of the aromatic chemical mechanisms within the MCM.

*The major results are presented primarily as figures giving fractions of ion signals that have various characteristics, plus some summary information given in the text. The paper has a "Supplementary Data" (SD) section to give additional information, and ideally it should have the information that is summarized or shown*

*graphically in the text, so the reader can examine it in more detail, to either to verify the discussion in the paper or perhaps to gain other insights. The SD does have 20 tables giving the "top 20" product distributions for each of the 2 types of experiments with 2 analysis methods and 5 compounds, but it only has the information regarding the ion detected and true/false flags indicating whether it is common or unique among the compounds studied and whether it has the same molecular formula as a product predicted by MCM. That is not the most interesting information they could present. These tables should include at least the relative ion signal intensity, and ideally also the classifications as ring-scission, ring-retaining, HOM, DBE, and other classifications they discussed or summarized. Instead of just indicating that this may be predicted by MCM, they should give the name and structure of the products(s) corresponding to this molecular weight. This would make the tables much more interesting and greatly increase the value of this work and information content of the paper.*

> Tables with this additional information have been included in a revised supplement. Structures have not, however been included as we feel these may be misleading as the mass spectrometry techniques used in this study cannot distinguish structures.

*The discussion of MCM and mechanistic implications could be improved. It is not surprising that MCM does not predict the full range of products they observe, especially HOM, because (1) the version of MCM that is currently online does not have autooxidation reactions that are now believed to be important, and (2) it employs lumping or reduction methods when it gets to 3+ generation products of the compounds represented. What might be more interesting would be high yield products that MCM predicts that they DO NOT observe. These should be listed, or it should be stated that there are no such products if that is in fact observed. One way to do this would be to run MCM to model the conditions of the experiments, summarize the yields predicted, and list these against the observed relative ion signals of products with the same molecular weight in the experiments. Are the products observed more consistent with the revised mechanisms predicted by Wang et al (2017)?*

> The referee provides suggestions of how our discussion of MCM and mechanistic implications can be improved. We agree that it is not surprising that the MCM does not predict the full range of products, especially HOM and describe the comparison with MCM as a benchmark against which observations can be compared in 4.1.1. As the MCM itself was not run for these simulations, owing to the objectives of this study not being to describe yields but instead product distributions, the high products cannot be distinguished. We have, however include a list of the ions within the MCM that are not observed in these experiments in the supplement as suggested.

> The closed shell products of $C_9H_{12}O_6$ and $C_9H_{12}O_8$ observed in Wang et al. 2017 are consistent with our observations from the aromatic systems. We have added this clarification in section 4.2.2.

*The tables in the SD indicate that no signals were observed for glyoxal ($C_2H_2O_2$), which is known to be formed in significant yields from all products (and is predicted by MCM). Also, methyl glyoxal ($C_3H_4O_2$) should also be seen as a product from the trimethylbenzenes and propyl glyoxal ($C_5H_8O_2$) should be formed from propyl benzene. Does this method not work for alpha-dicarbonyls? If so, state this. Or are they all rapidly photolyzed away by the high UV in their system? This gives me some doubt about the credibility of the experimental system.*

> Glyoxal explanation added:

"It should be noted that glyoxal, a major product from aromatic oxidation is detected by the Vocus as $C_3H_3O_2^+$ which is dominated in these experiments by protonated acetone. The sensitivity of glyoxal in PTR is almost one tenth of that of acetone, and furthermore C3H3O2+ can also be influenced by fragmentation, and thus is not included in the analysis herein (Pang et al., 2014; Stönner, Derstroff, Klüpfel, Crowley, & Williams, 2017; Thalman et al., 2015)."

Methylglyoxal explanation added:

"It should be noted that this mass range omits methyl glyoxal, a dominant product of aromatic oxidation, which was observed by Vocus in all experiments."

Propyl glyoxal is observed and already exists within the supplement of the paper as detected, except for propyl benzene in which it is observed but is not within the Top 20 dominant ions.

*It is stated that they did not correct for sensitivity of the ion signals to the actual concentrations in their analyses, apparently because of the complexities involved. However, since they are using the ion signal as a surrogate for the yields, it would be useful to estimate the uncertainty or potential for error when not making this correction.*

As previously outlined, we are not using the ion signal as surrogate for yields, as is suggested by the referee, and thus we cannot put uncertainty estimates upon a relationship we are not defining. We are aware of the significant uncertainties in relating ion signal to yields and thus have taken the approach of comparing product distributions in this manuscript. Thus, we deem it not necessary to estimate the error when correcting for sensitivities, when this data is not presenting the results as quantitative yields.

We have added a statement to the methodology to clarify this misunderstanding:

"It should be noted that we have not attempted to relate ion signals to yields and thus the comparison of ion signals in this study should not be assumed to be related to quantitative product yields derived from models such as those implementing mechanisms including the MCM. "

**Other Comments**

*The abstract should state what they mean by "product signal" and describe the analysis methods in a few words.*

Added:

"A time-of-flight chemical ionisation mass spectrometer (ToF-CIMS) with iodide-anion ionisation was used with a filter inlet for gases and aerosols (FIGAERO) for detection of products in the particle phase, while a Vocus proton transfer reaction mass spectrometer (Vocus-PTR-MS) was used for detection of products in the gas phase. The signal of product ions observed in the mass spectra were compared for the different precursors and experimental conditions. "

*The statement in the abstract that the MCM "highlights" missing product pathways, but it is not clear from that statement if it is the MCM that is missing something (which is the case), or the experiments aren't finding something that MCM predicts (which may or may not be the case – see comments above – but is probably not what the authors meant to say).*

Rephrased:

"Ions corresponding to products contained in the near explicit gas phase Master Chemical Mechanism (MCMv3.3.1) are utilised as a benchmark of current scientific understanding, and comparison of these with observations shows that the MCM is missing a range of highly oxidised products from its mechanism."

*The observation that "A large proportion of the ring scission products observed in the particle phase are more oxidised than those previously reported" is significant and should be included in the abstract.*

Thank you for noting this. The text in the abstract has been rephrased to include:

"In the particle phase, the bulk of product signal from all precursors comes from ring scission ions, a large proportion of which are more oxidised than previously reported and have undergone further oxidation to form highly oxygenated organic molecules (HOM)."

*Was there only a single experiment of each of the 10 types listed on Table 1, or are the concentrations given there the averages of several experiments. If the latter, indicate the number of experiments?*

Rephrased:

"VOC concentrations during single experiments for each precursor"

*The legend on the top right on Figure 1 should be moved to where it doesn't get in the way of the data being shown.*

[Figure]

*The labels indicating the compounds on Figure 3 need to be larger and easier to read. Right now they are in a very tiny font hugging the axes and are hard to see.*

These have been amended.

*Could they use a different color on Figure 3 to indicate N-containing ions?*

As we state on lines 308-310 of the revised manuscript, specific N-containing ions have not been included in further analysis due to potential contributions from thermal decomposition, thus we feel it is not relevant to include these in the figure.

*Table 2 would have more impact if it were shown as a bar-plot figure. It clearly shows n-propyl benzene as an outlier among the alkylbenzenes in terms of NOx effects. (I would have thought isopropyl benzene would be more likely the outlier, since the autooxidation proposed by Wang et al (2017) may be more important for this compound.) It might be useful to also include their estimates of ring-retaining and ring-scission fractions on this same table or figure, since there are also discussed in the text.*

We would like to thank the reviewer for bringing our attention to this insightful figure, since not only has the new figure improved clarity, it has enabled us to find an error in the Table itself for the HOM contribution in the 1,3,5-trimethyl benzene experiments. This does not influence the conclusions of the manuscript itself as HOM from 1,3,5-trimethyl benzene was high in both the table and in the Figure now presented. An improved figure has now been included, which improves the mechanistic insights through combining the ring-retaining and ring-scission classifications with those of HOM and non-HOM to enable all key features of the systems to be observed. The detailed proportions have been amended accordingly through the text.

[Figure]

*It is not clear what is the purpose of table 3, giving MCM branching ratios for various types of reactions, since they give no comparison of this with their data. If they can identify their products by the classifications on this*

*table, why don't they give the relative amounts derived from their data? If not, why include this table and discussion?*

> This table was presented to provide a context for the mechanistic discussions, however has now been removed.

The "Authors Contributions" section does not list all the authors.

> As we have made clear in the author contributions, all authors contributed to discussions of the results and comments on the manuscript. We have included explicit reference where people had explicit role. We hope this is acceptable. We can list all authors under the contribution to the manuscript but we feel this would be less clear to read.

The supplement document does not identify the title, authors, and journal of the manuscript it goes with.

> This has been added into the supplement.

**Anonymous Referee #2**

> We thank the referee for their comments which we have used to improve the manuscript. As is stated above, the main comments are relating to a misunderstanding that the paper was comparing yields derived from the MCM, and this has been clarified in the paper to address this. Below we address the referee's specific comments.

*Archit Mehra and co-workers conducted the evaluation of the chemical composition of gas and particle phase products of aromatic oxidation. Gas and particle phase composition are compared with the simulation results by Master Chemical Mechanism (MCMv3.3.1). This work highlights a series of missing highly oxidized products in the pathways. The work is therefore valuable in this regard. The article can be published once the authors have addressed the following points.*

*1. Line 31: "HOMs" Abbreviations should be given their full names when they first appear. Also, for Line 47 "SAPRC ".*

> Move full names from line 33 to line 31.

> "the Statewide Air Pollution Research Center of the University of California in Riverside (SAPRC) mechanism" and (Carter, 1988) reference.

*2. For the experiment of 1-methylnaphthalene: why the mixing ratio of VOCs is lower than other experiments? And for each VOCs, the experiments were conducted only once? If not, in all the experiments the mixing ratio of VOCs was controlled uniformly?*

A lower VOC mixing ratio was required in this experiment to obtain appreciable SOA mass. The experiments were only conducted once.

Changed caption of Table 1:

"Table 1 - VOC concentrations during single experiments for each precursor"

Added explanation of different VOC concentrations:

"Their concentrations (Table 1) were optimised to obtain similar aerosol mass concentrations in the different experiments."

3. Line 141ïïjŽwhat is the "PFA"?

Line 141 Inserted: "perfluoroalkoxy alkane (PFA)"

4. Line 144: you should change the "N2" into "N2", and you should check the paper to avoid the similar error.

Line 144 Inserted: "$N_2$" subscripted. Checked all paper for N2 and replaced with "$N_2$"

5. Figure 2: What is the ordinate?

Insert Y-Axis Label: "Ion Counts"

6. Figure 3, Figure 4: The quality of images needs to be improved.

Made axis labels clearer and larger in line with reviewer 1's comments also.

7. Line 273: how to identify the C4H6O2 and C4H8O3? Give the spectrum information.

Added to supplement as Figures S6-S7.

[Figure]

Figure S6 – High resolution peak fit of the ion C4H7O3+ corresponding to C4H6O3 detected by Vocus

[Figure]

Figure S7– High resolution peak fit of the ion C4H9O3+ corresponding to C4H8O3 detected by Vocus

8. Line 277-278: "with a small reduction in the fraction of C4 product ions and an increase in C2 product signal" what is the reason?

The ion contributing to this has been added to the discussion:

"Under medium-$NO_x$ conditions, the dominant gas phase ions are consistent with those produced under low-$NO_x$ conditions and the product signal shows a broadly similar distribution to the medium-$NO_x$ conditions, with a small reduction in the fraction of $C_4$ product ions and an increase in $C_2$ product **signal which can be attributed to an increase in $C_2H_4O_3$.**

9. In this paper, are there any new mechanism that could optimize the MCM? When the missing highly oxidized products were considered in MCM, is there any difference between simulation and experiments?

> The reviewer here has interpreted that the conditions in the OFR were modelled using the MCM mechanisms, which as highlighted above was not the case. A sentence of additional clarification is added:
>
> "Note that models incorporating MCM chemistry were not run for any particular conditions for these comparisons, instead the potential products included in the detailed MCM aromatic chemical mechanisms were simply compared with the ions observed in this study."
>
> New mechanistic details which could optimise the MCM are described in a companion paper (Wang et al. 2020).  This has been referenced the paper:
>
> "A companion paper provides more detailed discussion of the mechanisms responsible for formation of the dominant oxidised products from these precursors (Wang et al., 2020) .